# Frontline Sodium Channel-Blocking Antiseizure Medicine Use Promotes Future Onset of Drug-Resistant Chronic Seizures

**DOI:** 10.3390/ijms24054848

**Published:** 2023-03-02

**Authors:** Dannielle Zierath, Stephanie Mizuno, Melissa Barker-Haliski

**Affiliations:** Department of Pharmacy, School of Pharmacy, University of Washington, Seattle, WA 98195, USA

**Keywords:** carbamazepine, levetiracetam, gabapentin, perampanel, valproic acid, phenobarbital, topiramate, neurogenesis, Ki-67

## Abstract

The mechanisms of treatment-resistant epilepsy remain unclear. We have previously shown that frontline administration of therapeutic doses of lamotrigine (LTG), which preferentially inhibits the fast-inactivation state of sodium channels, during corneal kindling of mice promotes cross-resistance to several other antiseizure medicines (ASMs). However, whether this phenomenon extends to monotherapy with ASMs that stabilize the slow inactivation state of sodium channels is unknown. Therefore, this study assessed whether lacosamide (LCM) monotherapy during corneal kindling would promote future development of drug-resistant focal seizures in mice. Male CF-1 mice (*n* = 40/group; 18–25 g) were administered an anticonvulsant dose of LCM (4.5 mg/kg, i.p.), LTG (8.5 mg/kg, i.p.), or vehicle (0.5% methylcellulose) twice daily for two weeks during kindling. A subset of mice (*n* = 10/group) were euthanized one day after kindling for immunohistochemical assessment of astrogliosis, neurogenesis, and neuropathology. The dose-related antiseizure efficacy of distinct ASMs, including LTG, LCM, carbamazepine, levetiracetam, gabapentin, perampanel, valproic acid, phenobarbital, and topiramate, was then assessed in the remaining kindled mice. Neither LCM nor LTG administration prevented kindling: 29/39 vehicle-exposed mice were kindled; 33/40 LTG-exposed mice were kindled; and 31/40 LCM-exposed mice were kindled. Mice administered LCM or LTG during kindling became resistant to escalating doses of LCM, LTG, and carbamazepine. Perampanel, valproic acid, and phenobarbital were less potent in LTG- and LCM-kindled mice, whereas levetiracetam and gabapentin retained equivalent potency across groups. Notable differences in reactive gliosis and neurogenesis were also appreciated. This study indicates that early, repeated administration of sodium channel-blocking ASMs, regardless of inactivation state preference, promotes pharmacoresistant chronic seizures. Inappropriate ASM monotherapy in newly diagnosed epilepsy may thus be one driver of future drug resistance, with resistance being highly ASM class specific.

## 1. Introduction

The percentage of patients with pharmacoresistant epilepsy has remained unchanged despite the availability of over 30 antiseizure medicines (ASMs), yet the precise mechanisms underlying development and prevalence of treatment-resistant epilepsy are relatively unclear [1]. Understanding how frontline ASM monotherapy promotes the onset of future pharmacoresistant seizures would be of immense value to inform the appropriate selection of an ASM in an individual with newly diagnosed epilepsy. Further, this information would critically differentiate between currently available ASMs for frontline prescribing practice upon epilepsy diagnosis. Use of an inappropriate ASM monotherapy may adversely affect disease trajectory, but the randomized clinical trials to prospectively address this hypothesis have thus far not been completed, in part because of the limited preclinical evidence available to support such a complex hypothesis-driven study, as well as the ethical limitations associated with potentially imparting drug resistance on a person with epilepsy. Preclinical studies of the impact of initial ASM monotherapy selection on subsequent pharmacoresistance can thus guide rational ASM prescribing practice after clinical epilepsy diagnosis.

The traditional 60 Hz corneal kindled mouse (CKM) represents a well-characterized preclinical model of temporal lobe epilepsy (TLE) that has contributed to the identification and development of numerous ASMs [2,3], including levetiracetam [2]. The National Institute of Neurological Disorders and Stroke (NINDS) Epilepsy Therapy Screening Program (ETSP) has prioritized early drug screening in this model to incorporate more etiologically relevant epilepsy models earlier in the drug discovery process [4,5]. The 60 Hz CKM exhibits behavioral deficits [6,7,8] and pathophysiology consistent with human TLE (e.g., reactive gliosis in hippocampal structures [9]), thus making it useful to evaluate behavioral and pathophysiological changes associated with chronic seizures. The kindling model is unique in that it can generate many mice with uniform seizure and drug exposure history for subsequent experimental manipulation. Therefore, this model is incredibly valuable to explore pharmacotherapeutic drivers of pharmacoresistance in early epileptogenesis and define whether specific frontline ASMs may induce future drug-resistant epilepsy. 

Clinical guidance recommends frontline use of lamotrigine (LTG) and lacosamide (LCM) in newly diagnosed epilepsy indications. We have previously extensively characterized the pharmacological and behavioral profile of the LTG-resistant 60 Hz CKM based on the larger, more resource-intensive LTG-resistant amygdala-kindled rat [10,11]. Chronic administration of anticonvulsant doses of LTG during kindling acquisition, essentially the epileptogenesis period [12,13], does not delay acquisition of the fully kindled state but instead leads to a kindled mouse with marked resistance to retigabine, carbamazepine, and valproic acid, reflective of a suitable drug-resistant epilepsy model [8]. Inappropriate ASM monotherapy early in the disease course may thus drastically influence the onset of future pharmacosensitivity; a finding that is reported in monozygotic twins with epilepsy [14] and also in resected brain tissues from people with drug-resistant TLE [15]. The LTG-resistant CKM is thus an appropriate preclinical model to interrogate the factors that drive the development of pharmacoresistant epilepsy, as well as the sequelae associated with pharmacoresistant TLE. Further, CKM facilitates complex drug efficacy tests with a reduced demand on animal welfare, addressing a major objective of the 3Rs in animal research [16].

It is currently unclear whether the development of drug resistance in the available preclinical models [8,10,11] is due to sodium channel-blocking ASM administration, or a function of LTG monotherapy alone. LTG is mechanistically similar to other sodium channel-blocking ASMs, i.e., carbamazepine and phenytoin, in that it preferentially inhibits the fast inactivation state, whereas LCM is an ASM broadly residing in the sodium channel-blocking class, but dissimilar in that it preferentially stabilizes the slow inactivation state and conformational change of sodium channels [17]. We thus hypothesized that frontline monotherapy with inactivation state-specific sodium channel-blocking ASMs would not impact the subsequent development of pharmacoresistance of the corneal kindled seizure, reflecting a class-specific effect of chronic sodium channel modulation rather than an effect isolated to LTG administration alone. This study therefore provides novel preclinical evidence that inappropriate selection of sodium channel blocking ASMs in early epileptogenesis can by itself promote the future development of drug-resistant epilepsy. 

## 2. Results

### 2.1. Administration of Neither LTG nor LCM at Anticonvulsant Doses Delays Kindling Acquistion

We have previously demonstrated that administration of an anticonvulsant LTG dose does not delay kindling acquisition in male CF-1 mice [8]. LCM has been found to delay amygdala-kindling acquisition in male rats in a dose-related manner, but the high dose that delayed kindling also exerted significant motor impairment [18]. Therefore, our first objective was to define the median anticonvulsant dose of LCM in male CF-1 mice in the maximal electroshock test (MES; Table 1) to define the anticonvulsant dose needed to match our earlier studies with LTG [8]. We quantified the i.p. median effective dose (ED50) of LCM in seizure-naïve male CF-1 mice from Envigo as 4.05 mg/kg [3.65–4.80; Table 1], consistent with the previously reported ED50 for this agent in CF-1 mice from Charles River [19]. Therefore, a 4.5 mg/kg (i.p.) dose was subsequently acutely administered to a new cohort of CF-1 mice prior to each twice-daily corneal stimulation session to define whether an anticonvulsant dose of LCM would similarly delay corneal kindling (Figure 1).

### 2.2. Administration of Either LTG or LCM at Anticonvulsant Doses during Kindling Reduces Subsequent Kindled Seizure Cross-Sensitivity 

We have earlier shown that administration of an anticonvulsant dose of LTG twice a day during corneal kindling of male CF-1 mice does not delay kindling and instead leads to the subsequent development of a highly drug-resistant kindled seizure [8]. We thus tested the hypothesis that cross-tolerance to other ASMs would arise in mice similarly exposed to LCM. We quantified the seizure-suppressive effects of a diversity of ASMs in mice that received either LTG or LCM during the kindling process, to determine whether this observation could similarly extend to mice exposed to LCM (Figure 2). The doses selected for the various ASMs matched those that have been previously evaluated in LTG-resistant CKM and previously published ED50s for these agents in a diversity of mouse focal seizure models [4,8,20]. 

Following a 5–7-day- seizure-free period, fully kindled mice within each kindling treatment group were first challenged to determine their sensitivity to a doubling of the initially exposed dose of each sodium channel-blocking ASM (i.e., 17 mg/kg for LTG and 9 mg/kg for LCM), matching our prior methods in CKM exposed to LTG [20], as well as approaching the previously determined median behaviorally impairing dose for each agent (50 mg/kg for LTG [20] and 26.5 mg/kg for LCM [19,21]). This strategy reflects a dose-escalation approach in a clinical setting for treatment-resistant epilepsy. 

Acute administration of LTG to fully kindled mice conferred a drug dose x kindling group interaction on mean seizure score (F_(4, 42)_ = 2.635; *p* = 0.0473; Figure 2A). Within kindling groups, LTG exerted a dose-related anticonvulsant protective effect in VEH-kindled mice (Table 2), and high-dose LTG significantly reduced mean seizure score in these mice (*p* < 0.0001). However, LTG administration to LTG-kindled mice led to a significant reduction in the total number of mice that were protected from seizure at the highest dose administered (50 mg/kg; Table 2), although there was a small, but significant, reduction in mean seizure score (Figure 2A; *p* = 0.048). Similarly, administration of escalating doses of LCM to LTG-kindled mice led to a significant reduction in the total number of mice that were protected from seizure at the highest dose administered (Table 2), but there was no significant reduction in mean seizure score (Figure 2). This demonstrates that cross-tolerance to LCM occurs following LTG exposure during early epileptogenesis.

Analogously, acute administration of LCM to fully kindled mice conferred a significant main effect of drug administration on mean seizure score (F_(2, 41)_ = 16.58; *p* < 0.0001; Figure 2B). LCM exerted a dose-related anticonvulsant effect in VEH-kindled mice (Table 2) and LCM significantly reduced mean seizure score in these mice in a dose-related manner (*p* < 0.01 at both 9 and 26.5 mg/kg). However, LCM administration to LTG-kindled mice did not significantly reduce the total number of mice that were protected from seizure at either dose administered (Table 2), although there was a small, but significant, reduction in mean seizure score at the highest dose of LTG administered to LCM-kindled mice (Figure 2B; *p* = 0.018). Similarly, administration of two doses of LCM to LCM-kindled mice did not significantly reduce the total number of mice that were protected from seizure at either dose administered (Table 2) and there was no significant reduction in mean seizure score (Figure 2B). These data illustrate that early exposure to therapeutic doses of the ASMs LTG or LCM ultimately leads to reduced sensitivity to dose escalation of the same ASM, as well as significant cross-tolerance to the alternative agent (i.e., LTG switched to LCM is ineffective). 

### 2.3. Administration of Either LTG or LCM at Anticonvulsant Doses during Kindling Leads to Significant Drug Resistance across a Range of ASMs with Diverse Mechanims

Patients with epilepsy who fail their first ASM are transitioned to an alternative agent but there is often little consideration for the mechanism of action of the following or add-on treatment. Therefore, we wanted to serially determine whether the efficacy of any ASM class was particularly subsequently impacted in LTG- or LCM-kindled mice following confirmation of the lost sensitivity to these ASMs (Figure 3). The resulting profile of the selected ASMs tested in LTG- and LCM-kindled mice can be classified into three principal profiles: no change in potency; loss of potency; and loss of efficacy (Figure 3 and Table 2). 

Two of the ASMs tested demonstrated no change in potency across the three kindling groups. Levetiracetam (LEV) administration at escalating doses was associated with a significant main dose effect on mean seizure score (F_(2, 64)_ = 83.46, *p* < 0.0001), and post-hoc analysis indicated that regardless of kindling group, the low and high doses of LEV resulted in significant reductions in mean seizure score (*p* < 0.0001 for groups; Figure 3A). Further, protection scores did not significantly differ between treatment groups (Table 2). Similarly acute administration of gabapentin (GBP) was associated with a significant main effect of ASM dose (F_(2, 66)_ = 24.68, *p* < 0.0001). While low-dose GBP did not significantly reduce mean seizure score (Figure 3B) or protect mice from seizures (Table 2), there was a significant reduction in mean seizure score and total number of protected mice with high-dose GBP administration, regardless of kindling group (Figure 3B, *p* < 0.01 for all). Further, high-dose GBP led to a significant increase in the number of mice protected from seizures in all kindling groups (Table 2; *p* < 0.01 for all). Thus, animals that were kindled in the presence of anticonvulsant doses of LTG or LCM remained sensitive to escalating doses of LEV and GBP. 

Several other ASMs tested demonstrated reduced potency to block the fully kindled seizure in the LTG- and LCM-kindled mice. Specifically, administration of low-dose perampanel (PER) significantly reduced mean seizure score (Figure 3C; F_(2, 66)_ = 79.50, *p* < 0.0001), and post-hoc tests indicated that VEH-treated mice had significant score reductions at both the low and high doses tested (*p* < 0.0001). PER administration also significantly increased the number of mice protected from seizure in VEH-kindled mice (Table 2). However, only high-dose PER significantly reduced mean seizure score in LTG- and LCM-kindled mice (Figure 3C; *p* < 0.001 for all) and significantly increased protection from seizures (Table 2). Similarly, valproic acid (VPA) conferred a main dose effect (F_(2, 42)_ = 27.57, *p* < 0.0001), and post-hoc analysis demonstrated that VEH-kindled mice had dose-related reductions in mean seizure score following VEH administration (Figure 3D). There was also a significant dose-related increase in the number of protected mice (Table 2). VPA, however, lost potency in LTG- and LCM-kindled mice; only the 300 mg/kg dose reduced mean seizure score (Figure 3) and increased the proportion of protected mice (Table 2). Finally, phenobarbital (PB) also significantly reduced mean seizure score (Figure 3E), revealing a significant effect of drug dose (F_(2, 42)_ = 40.22, *p* < 0.0001), with post-hoc analysis revealing dose-related reductions in mean seizure score (Figure 3E) in VEH-kindled mice. PB also significantly increased the number of protected mice at both doses tested (Table 2). Yet there was a loss of potency in LTG- and LCM-kindled mice; only the 45 mg/kg PB dose reduced mean seizure score (Figure 3; *p* < 0.001) and increased the proportion of protected mice (Table 2). Therefore, these three mechanistically distinct ASMs lost potency in mice kindled in the presence of therapeutic doses of LTG or LCM.

Exposure to LCM during the kindling process led to cross-resistance to escalating doses of carbamazepine (CBZ; Figure 3F and Table 2). Like LTG and LCM, CBZ is another sodium channel-blocking ASM, and we have previously demonstrated that neither CBZ nor phenytoin retain potency in LTG-exposed CKM [8]. This is an effect that is similar to the effect seen in LTG-resistant amygdala-kindled rats [10,11]. There was a main dose effect in all treatment groups (F_(2, 42)_ = 24.07; *p* < 0.0001; Figure 3F), with VEH-kindled mice demonstrating significant reductions in seizure score at both doses tested (*p* < 0.01). However, post-hoc tests demonstrated that mean seizure score in LTG-kindled mice was not significantly reduced at either dose tested. Conversely, mice kindled in the presence of LCM only demonstrated a modest, albeit statistically significant, reduction in mean seizure score at the highest CBZ dose tested (*p* < 0.05). This did not, however, significantly impact the total number of protected mice in either LTG- or LCM-kindled mice. Further, topiramate (TPM), which has a mixed mechanism of action [17] and has not been previously reported to be effective in CKM [22], was not effective in any of the kindling groups at any dose tested (Figure 3G and Table 2).

### 2.4. LTG Administration during Corneal Kindling Blunts Chronic Seizure-Induced Increases in Reactive Gliosis in Area CA1 of Dorsal Hippocampus

Our earlier studies demonstrated that chronic kindled seizures promote increased astrogliosis as early as one day after achieving the fully kindled state, and that this increase is not due to astroglial proliferation [9,23]. Thus, we sought to assess whether LTG- or LCM-administration during the kindling process would impact glial fibrillary acid protein (GFAP) immunoreactivity at this same time point (Figure 4). GFAP is a marker of astroglial reactivity that is elevated in response to chronic seizure activity [9,23]. Consistent with our prior reports [9], there were significant kindling-induced increases in GFAP immunoreactivity in area CA1 (Figure 4A; F = 4.13, *p* = 0.016), but no other region exhibited reactive gliosis (Figure 4B, C). Further, post-hoc analysis demonstrated significant increases in GFAP immunoreactivity in VEH-kindled mice (*p* = 0.022) and LCM-kindled mice (*p* = 0.012) in area CA1. However, there was no significant difference from sham in LTG-kindled mice in this brain region. Thus, treatment with LTG, but not LCM, may blunt chronic seizure-induced increases in reactive gliosis in area CA1 of the dorsal hippocampus. 

### 2.5. LCM Administration during Corneal Kindling Blunts Chronic Seizure-Induced Increases in Neuronal Density in Dorsal Hippocampus

We also wanted to determine whether anticonvulsant administration of LTG or LCM during corneal kindling could influence neuronal density, as assessed by NeuN immunoreactivity in dorsal hippocampal structures (Figure 5), which would potentially suggest possible changes in neuropathology or neurodegeneration. We observed significant seizure-induced increases in NeuN immunoreactivity in area CA1 (F = 3.108, *p* = 0.042; Figure 5A) and dentate gyrus (F = 5.694, *p* = 0.004; Figure 5C). There was no significant effect of kindling on NeuN levels in CA3 (Figure 5B). Post-hoc analysis in CA1 revealed that only VEH- and LTG-kindled mice had significant increases in total field area with NeuN immunoreactivity (*p* < 0.05), an effect that was also evident in dentate gyrus (*p* < 0.01). Thus, NeuN immunoreactivity was significantly increased in area CA1 and dentate gyrus in VEH- and LTG-kindled mice only.

### 2.6. LCM Administration during Corneal Kindling Is Associated with Increased Neurogenesis in Dentate Gyrus

Considering the findings of increased NeuN immunoreactivity in VEH- and LTG-kindled mice, we sought to determine whether this increase was potentially the result of neurogenesis as measured by the molecular marker of cellular proliferation, Ki-67 (Figure 6) [24]. There was a significant main effect of kindling on the number of colocalized Ki-67+ and NeuN+ cells (F = 5.848, *p* = 0.003) within the dentate gyrus only; no other region of dorsal hippocampus demonstrated significant Ki-67+ labeling (not shown). However, post-hoc analysis revealed a significant increase in the number of Ki-67 + /NeuN+ colocalized cells only in mice kindled in the presence of LCM (Figure 6A). Thus, only LCM administration during corneal kindling significantly influenced dentate gyrus neurogenesis.

## 3. Discussion

The percentage of patients with drug-resistant epilepsy has remained unchanged for more than 20 years, despite the availability of over 30 ASMs [25,26]; there is thus a high unmet need for agents that are effective in individuals with pharmacoresistant epilepsy. Further, understanding how frontline ASM selection in early epileptogenesis affects the likelihood to ultimately develop pharmacoresistant epilepsy would be of immense value to appropriately select an initial ASM in the clinical setting. Such insight could also differentiate currently available ASMs for use in patients with newly diagnosed epilepsy. This present study confirms that exposure to sodium channel-blocking ASMs, regardless of inactivation state preference, during corneal kindling leads to a highly drug-resistant chronic seizure model. This drug-resistant seizure model is both resistant to subsequent dose escalation with the same agent, as well as to cross-over of mechanistically related ASMs (i.e., LTG, LCM, and CBZ are all sodium channel-blocking ASMs). Furthermore, we presently reveal stark differences in neuroinflammation and neurogenesis between mice exposed to anticonvulsant doses of LTG versus LCM during kindling, suggesting that ASM monotherapy during kindling can dramatically alter the resulting neuropathology with chronic seizures. These findings extend our earlier work to indicate that exposure of mice to LTG during corneal kindling induces a state of subsequent pharmacoresistance [8], which closely aligns with findings following chronic administration of LTG to either amygdala-kindled rats [10,11] or pentylenetetrazol-kindled mice [27]. Importantly, we now demonstrate that there is a significant loss of potency with the ASMs VPA, PB, and PER, or an altogether loss of anticonvulsant efficacy with mechanistically related agents (in the case of CBZ). This work clearly indicates that inappropriate ASM monotherapy selection alone can dramatically alter later ASM sensitivity and likely instead lays the foundation for future drug-resistant epilepsy. Our work carries substantial translational impact for the management of treatment-resistant epilepsy. Further, this study suggests a possible point of clinical caution in the frontline selection of an ASM regimen for a newly diagnosed patient with epilepsy. Our present findings suggest that inappropriate ASM selection in early epileptogenesis could be one contributing mechanism to influence the subsequent onset of treatment-resistant epilepsy. While many contributing factors have been proposed to drive the development of pharmacoresistant epilepsy [1], the idea that ASM monotherapy alone may in any way contribute to the ultimate acquisition of drug-resistant epilepsy has been less extensively clinically investigated.

Treatment-resistant epilepsy likely arises due to several inciting mechanisms [1], including modifications of therapeutic targets [28], changes in multidrug transporters at the blood brain barrier (BBB) [29], intrinsic differences in seizure severity [30], and even failures in medication adherence [31]. Although we did not exclusively evaluate sodium channel subunit composition, function, or density in this study, it is possible that the subsequent treatment resistance observed in both LTG- and LCM-kindled mice may arise because of modifications in sodium channels themselves. LTG and CBZ share the same binding site [32] and high affinity for the open/fast inactivated state of the sodium channel [33], although LCM preferentially stabilizes the slow inactivation state conformation of sodium channels [34,35,36], suggesting that a global modification of the sodium channel availability or function underlies the presently observed treatment-resistant chronic seizure model. Consistent with our earlier findings in the LTG-resistant CKM [8], mice exposed to LCM in this study also are resistant to PER, PB, and VPA, and altogether lose sensitivity to CBZ (Figure 3 and Table 2). CBZ, LTG, and VPA all exert some anticonvulsant effects through sodium channels [17,37]. PHT shares the same binding site on neuronal sodium channels as LTG and CBZ [32,33]; however, we did not test the activity of this agent in LCM-kindled mice, because we have already documented lack of activity with this agent in LTG-resistant CKM [8]. Clinically, there is a strong indication that mutations in the Nav1.1 subunit of sodium channels are heavily involved in treatment-resistant seizures (i.e., Dravet syndrome [38]). Additionally, sodium channels and KCNQ potassium channels can co-localize and are positioned in proximity within the neuronal membrane [39,40], with KCNQ2 encephalopathy representing another highly treatment-resistant epilepsy syndrome [41]. Whether potassium channel activators would be ineffective in the LCM-resistant CKM, as we have previously reported with ezogabine administration in LTG-resistant CKM [8], requires future studies, but would further bolster the hypothesis that modifications in the availability or intrinsic properties of sodium channels specifically evoke onset of drug-resistant epilepsy. It is also possible that chronic blockade of sodium channels can evoke functional changes in potassium channel function, as a similar phenomenon occurs following chronic exposure to the potassium channel blocker, dofetilide, in cardiomyocytes and leads to increased late sodium currents [42]. Thus, additional investigation into the bidirectional functional changes in sodium and potassium channel expression and function after chronic exposure to LCM and LTG during epileptogenesis are clearly necessary. 

Alternative contributors to treatment-resistant epilepsy may also underlie the presently reported shift in ASM efficacy in mice with a history of LTG or LCM exposure during kindling. Efflux transporter changes at the BBB, e.g., P-glycoprotein (PgP), may also promote pharmacoresistance [28]. The PgP inhibitor, tariquidar, has successfully rescued drug sensitivity in PB-resistant rats with epilepsy [29], directly supporting the transporter hypothesis. PB is a PgP substrate [29,43], as are LTG and LEV [43], but only PB and LTG efficacy were significantly diminished following LCM or LTG administration during kindling. LEV retained efficacy in this present study, as well as in our previously published work with the LTG-resistant CKM [8]. CBZ is also not a substrate for PgP [43]. Thus, our present and prior studies clearly rule out changes in drug transporters as a major contributor to drug resistance in this model, and instead strongly point to changes in sodium channels themselves as a key factor promoting ASM resistance.

One core goal of this study was to define how LTG versus LCM monotherapy changes hippocampal neuropathology and neurogenesis at a cellular level. We found that LTG or LCM as a frontline monotherapy led to substantial changes in cellular and behavioral outcomes. One novel finding of our present study is the observation of differential impacts on astroglia immunoreactivity in area CA1 of dorsal hippocampus following LTG versus LCM administration during corneal kindling (Figure 4A). LTG administration did not lead to marked changes in astrocyte immunoreactivity relative to sham-treated mice, whereas both VEH- and LCM-treated mice exhibited significant astrogliosis in this brain region. Moreover, the density of NeuN immunoreactivity, a marker of mature, post-mitotic neurons [44], was only significantly increased versus sham-kindled mice in VEH- and LTG-treated mice in areas CA1 and dentate gyrus of the dorsal hippocampus (Figure 5A,C). LCM-kindled mice did not demonstrate significant increases in NeuN immunoreactivity in any brain region relative to sham-kindled mice (Figure 5). We then assessed whether these changes in NeuN immunoreactivity were due to changes in neurogenesis within dentate gyrus using the protein marker Ki-67. This protein is expressed in all phases of the cell cycle except the resting phase and is a suitable marker of mitosis equivalent with BrdU [24]. Ki-67-positive/NeuN-positive cell counts were only robustly upregulated in dentate gyrus with LCM administration (Figure 6). Repeated kindled seizures induce premature stem cell differentiation and neurogenesis [45], which may itself further drive mossy fiber sprouting and formation of an epileptic network. Considering that our limited dose-response study was not statistically powered to quantify an ED50 of each ASM in LCM- versus LTG-treated CKM, we cannot presently conclude whether either ASM monotherapy led to any differential ASM potency. Instead, the present study demonstrates that ASM monotherapy alone can strikingly influence the resulting neuropathology, which warrants further detailed study. Voltage-gated sodium channels may promote T-cell development [46], activation [47], and invasiveness [48], further suggesting that neuroinflammation may have been chronically differentially modulated by the LTG versus LCM administration. Future studies are thus needed to determine the mechanistic basis of this difference.

From a clinical perspective, our present findings need additional in-depth analysis in a real-world setting to determine whether initial ASM monotherapy influences subsequent treatment continuity or treatment switching, indicative of drug-resistant epilepsy Inappropriate ASM monotherapy early in the disease course may drastically alter the trajectory of future pharmacosensitivity, as has been observed in a clinical case report from monozygotic twins [14]. However, few studies have been completed to determine real-world ASM prescribing practices and the subsequent impact of ASM monotherapy on patient outcomes. Faught et al. previously reported that within a population of 12,975 US patients between 2010–2013, LEV was the frontline ASM of choice in 44.4% of all individuals, regardless of seizure type [49], but sodium channel-blocking ASMs, including PHT, LTG, and oxcarbazepine (OXC, a CBZ analogue), represented the next most common choice (18.5% of patients). When patients were stratified by epilepsy diagnosis (focal versus generalized) and insurance type (commercial and Medicare claims versus Medicaid), the ASM use distribution in focal epilepsy patients shifted to 49.7% (commercial/Medicare) and 42.6% (Medicaid) of patients receiving frontline LEV versus 23.8% (commercial/Medicare) and 23.5% (Medicaid) of patients receiving PHT, LTG, and OXC [49]. Conversely, the distribution of generalized epilepsy distribution was 46.0% (commercial/Medicare) and 39.1% (Medicaid) of patients receiving frontline LEV, versus 17.4% (commercial/Medicare) and 14.5% (Medicaid) of patients receiving PHT, LTG, and OXC. Interestingly, that earlier study demonstrated that patients who initially received PHT were the least likely to stay on this monotherapy regimen and the most likely to switch to another ASM treatment option [49]. The retrospective clinical study suggests that while sodium channel-blocking ASMs are not the most commonly prescribed frontline agent in newly diagnosed epilepsy, the initial prescribing of sodium channel-blocking ASMs, regardless of epilepsy diagnosis, is likely to be poorly tolerated and lead to ASMs switching [49], reflective of drug-resistant epilepsy. Our present study offers valuable insight to future clinical efforts to optimize ASM selection. Further, we justified the selection of LTG and LCM, two newer sodium channel-blocking ASMs, based on UK studies indicating that LTG and LEV prescribing has steadily increased whereas PHT has declined and CBZ has remained steady over the last several decades [50]. This retrospective study of over 63,586 UK patients indicates that sodium channel-blocking ASMs accounted for over 72% of all prescriptions between 1993–2008 [50]. These clinical studies, in consort with our present preclinical work, indicate that sodium channel-blocking ASMs are commonly prescribed in clinical practice, yet if seizure control is not successfully attained with an agent of this ASM class, the patient may be set up for future onset of treatment-resistant epilepsy. 

While this present work demonstrates some key, clinically relevant findings, it does not perfectly translate into clinical practice; the ASMs were administered acutely by the i.p. route (i.e., single dose) after a three-week course of twice-daily LTG or LCM administration during the epileptogenesis period. Further, we did not assess brain or plasma concentrations to know whether pharmacokinetic changes were in any way responsible for the present shift in anticonvulsant potency of PB, PER, or VPA. Nonetheless, the doses at which anticonvulsant efficacy was attained with VPA and PB approximate those that have been previously determined to be median behaviorally impairing doses for these agents [19,20], revealing a narrowing of the margin between seizure control and onset of adverse side effects in mice with a history of LTG or LCM exposure during corneal kindling. These data altogether highlight a possible opportunity to shift ASM monotherapy prescribing practice, or at least advise a cautionary assessment of risk/benefit relationship in the use of sodium channel-blocking ASMs in an individual with newly diagnosed epilepsy. Advancements in modern genetic sequencing technologies will be instrumental to more rationally guiding ASM selection and interventions for individuals with discrete genetic variants in sodium or potassium channel function [41], possibly ultimately resulting in a reduction in the percentage of patients with treatment-resistant epilepsy.

## 4. Materials and Methods

Animals: All animal experimentation was approved by the University of Washington Institutional Animal Care and Use Committee under approval number 4387-01 (MBH; approval date 5/5/2019), University of Washington Public Health Service (PHS) Assurance issued by the U.S. Office of Laboratory Animal Welfare (OLAW) assurance number D16-00292, and University of Washington AAALAC accreditation number #000523. Male CF-1 mice (4–8 weeks; Envigo Laboratories, Indianapolis, IN, USA) were housed, five mice/cage, in a temperature-controlled vivarium on a 14:10 light/dark cycle. Mice had free access to irradiated chow (Picolab 5053) and water, except during periods of behavioral seizure testing, as previously detailed [51]. Mice were allowed a minimum of four days’ habituation to the housing facility, given a minimum of 1 h to acclimate to the procedure room prior to all experimentation, and euthanized by decapitation at the completion of all in-life studies. Upon euthanasia at the designated time point after corneal kindling (or sham kindling), the brain was rapidly removed and the hippocampus isolated and flash-frozen on dry ice. Brain samples were stored at −80 °C until analytical processing. 

Maximal Electroshock Test (MES): A cohort of seizure-naïve mice (*n* = 63; Table 1) were used to define the ED50 of LCM in male CF-1 mice from Envigo, to confirm alignment with previous reports in CF-1 mice from Charles River [19] prior to embarking on any drug administration studies during the corneal kindling process. LCM was administered by the i.p. route 1 h prior to MES testing. For the MES test, 60 Hz of alternating current was delivered for 2 s by corneal electrodes. An electrolyte solution containing an anesthetic agent was applied to the eyes before stimulation (0.5% tetracaine HCl). The current intensity was 50 mA for mice. A mouse was considered “protected” from MES-induced seizures in the absence of the hindlimb tonic extension component of the seizure [4,52,53]. Quantification of the ED50 was conducted in groups of, at minimum, *n* = 8 mice, by administering escalating doses of LCM until at least two points could be clearly established between the limits of 0% and 100% protection. The ED50, the 95% confidence interval, the slope of the regression line, and the S.E.M. of the slope were calculated by Probit [54]. 

Corneal Kindled Mouse (CKM): For the 60 Hz corneal kindling protocol, mice (*n* = 40) were stimulated with an initially benign electrical current (60 Hz, sinusoidal pulse; 3.0 mA), delivered for 3 s via corneal electrodes or sham-kindled (*n* = 10). Seizures were scored on a 5-point rating scale consistent with the Racine scale in amygdala-kindled rats and routinely used by our group for corneal kindled mice [55], wherein; 1 = jaw chomping and vibrissae twitching, 2 = head bobbing and Straub tail, 3 = unilateral forelimb clonus, 4 = bilateral forelimb clonus and hind-limb rearing, 5 = bilateral forelimb clonus and rearing followed by loss of righting reflex. Twice daily stimulations continued for each mouse until it achieved the criterion of five consecutive Racine stage 5 seizures, whereby the mouse was considered “fully kindled”. Any mouse not achieving the fully kindled state was not included for further study. 

In Vivo Experimental Reagents: The following antiseizure medicines were commercially purchased: lamotrigine—AK Scientific, K499, (Union City, CA, USA); lacosamide—Cayman Chemical Co., 10012592, (Ann Arbor, MI, USA); carbamazepine—Sigma Aldrich, C4024, (Burlington, MA, USA); gabapentin—Tokyo Chemical Industry, G0318 (Tokyo, Japan); levetiracetam—Tokyo Chemical Industry, L0234 (Tokyo, Japan); perampanel—Cayman Chemical Co., 23003, ((Ann Arbor, MI, USA); topiramate—Tokyo Chemical Industry, T2755 (Tokyo, Japan); valproic acid—Sigma Aldrich, P4543. Tetracaine HCl (Sigma Aldrich, T3937) was formulated in 0.9% saline. All ASMs were formulated in 0.5% methylcellulose (Sigma Aldrich, M0430). Brains were rapidly collected following in vivo experimentation and flash frozen in 2-methylbutane overlying dry ice [56].

Antiseizure Medication Efficacy Studies in Fully Kindled Mice: Antiseizure medication (ASM) efficacy studies commenced at least 5–7 days after achieving kindling criterion. The same mice that were initially exposed to VEH, LTG, or LCM during the corneal kindling process and determined to be fully kindled were then used in a cross-over ASM screening design for up to two months after attaining the kindling criterion, matching our previously reported study design and ASM testing procedure [8,20]. Seizure scores of 2 or lower were considered “protected”. Mice were administered escalating doses of each ASM surrounding the known median effective dose (ED50) of each agent in male WT kindled mice at the previously determined time of peak pharmacodynamic effect [18,20]. Mice were allowed a minimum of three days’ washout between a dose of each ASM. ASMs were administered in a cross-over drug administration protocol to account for drug and seizure history [21].

Cryosectioning for Immunohistochemistry: Mice for histology were euthanized 24 h after meeting kindling criterion. Brains were flash frozen and maintained at −80 °C until sectioning on a cryostat (Leica DM1860). Four consecutive 20 μm-thick sections/mouse from the dorsal hippocampus (AP from Bregma: −1.58 to −2.18 mm) were slide-mounted on SuperFrost slides (Fisher). Slides were stored at −80 °C until immunohistochemical processing.

Immunohistochemistry: Brains of all mice were processed for quantitative assessment of molecular markers of reactive gliosis (GFAP; Sigma Aldrich C9205, St. Louis, MO, USA), neuronal integrity (NeuN; Millipore MAB377X, Burlington, MA, USA), and neurogenesis (Ki-67; Abcam ab16667, Tokyo, Japan) at the conclusion of corneal kindling (*n* = 9–10 mice/group). The GFAP/NeuN labeling followed our previously reported protocol [9,57].

Neurogenesis in hippocampal structures was assessed by Ki-67 labeling with a goat anti-rabbit secondary antibody conjugated to a 555 nm Cy3 fluorophore (Abcam ab150078, Tokyo, Japan). Briefly, slides labeled for Ki-67 were removed from −80 °C directly into fixative (4% paraformaldehyde; FD Neurotechnologies, Columbia, MD, USA) for 10 min. Slides were then washed 3 × 10 min in 1× PBS at room temperature (RT) before being incubated with a blocking reagent (10% goat serum with 0.1% Triton-X in 1× PBS) for 2 h at RT in a humid chamber. Slides were then incubated with the Ki-67 antibody (1:300) in 1% BSA in 1× PBS/0.1% Triton-X overnight at 4 °C in a humid chamber. The following day, slides were again washed 3 × 10 min in 1× PBS before incubation with the goat-anti rabbit secondary antibody (1:500) and NeuN (1:300) in 1× PBS/0.1% Triton-X for 1 h at RT in a humid chamber. Slides were then washed (3 × 10 min in 1× PBS) and mounted with ProlongGold with DAPI nuclear counterstain (Invitrogen P36935). Slides were allowed to cure for 24 h at RT before imaging.

Photomicrographs were captured with a fluorescent microscope (Leica DM-4) with a 20× objective (80× final magnification). Acquisition settings were held constant throughout. NeuN, GFAP, and Ki-67 expression levels, given as average area percentage, were automatically measured using Leica Thunder software. The number of Ki-67-positive cells were also hand-counted from all images of hippocampal structures (CA1, CA3, and dentate gyrus) by two investigators blinded to treatment, as well as by the automated cell counting feature of the Leica Thunder imaging software. Cell counts were averaged from all three reviewers and concordance determined by a coefficient of variance.

Statistics: The percentage of fully kindled mice was compared between experimental cohorts by an X^2^ test. The total numbers of stimulations needed to attain the fully kindled state between the treatment groups were compared by one-way ANOVA. Statistical differences in ASM response in fully kindled mice and immunohistochemical detection of protein expression were assessed by one-way ANOVA, with Dunnett’s post-hoc tests. For all statistical measures, *p* < 0.05 was considered significant and all analysis was performed with Prism version 8.0 or later (GraphPad, San Diego, CA, USA).

## Figures and Tables

**Figure 1 ijms-24-04848-f001:**
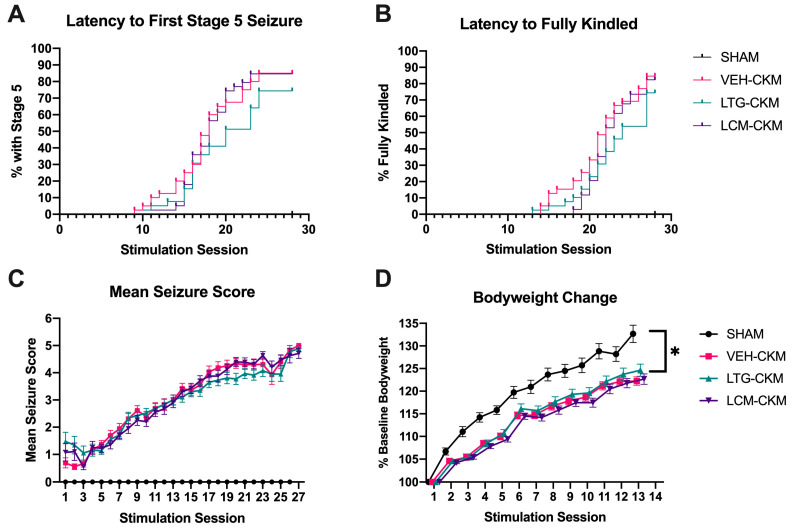
Repeated administration of neither lamotrigine (LTG—8.5 mg/kg, i.p.) nor lacosamide (LCM—4.5 mg/kg, i.p.) to male CF-1 mice at the respective time of peak effect prior to each twice-daily transcorneal 60 Hz stimulation over two weeks prevents the development of the fully corneal kindled state. (**A**,**B**) Electrical stimulation of the corneas at first does not evoke a behavioral seizure, but over the course of several days, the seizure severity becomes progressively more severe and the percentage of mice with a stage 5 seizure increases, (**B**) as does the percentage of mice that meet the kindling criterion of five consecutive Racine stage 5 seizures. There is no difference in proportion of fully kindled mice in either monotherapy group vs. VEH-kindled mice. (**C**) The mean seizure score is plotted for all treatment groups over time. (**D**) The change in body weight from baseline (kindling session 1) is generally blunted in kindled mice, but there is no significant treatment effect across kindled groups (* indicates *p* < 0.05).

**Figure 2 ijms-24-04848-f002:**
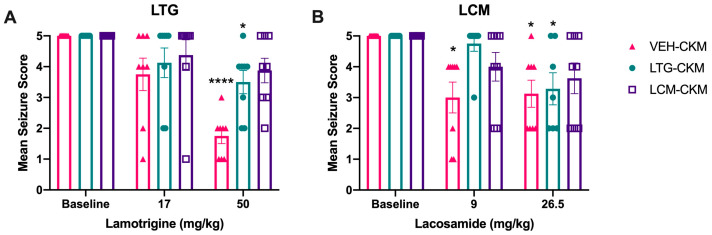
Repeated administration of anticonvulsant doses of either LTG (8.5 mg/kg) or LCM (4.5 mg/kg) to male CF-1 mice twice daily for two weeks during 60 Hz corneal kindling leads to cross-resistance to each sodium channel-blocking ASM at a subsequent doubling or behaviorally impairing dose of each agent (50 mg/kg, LTG; 26.5 mg/kg, LCM). Following a two-week kindling procedure, mice were allowed a 5–7-day stimulation free period before baseline seizure score was assessed (baseline) to confirm stable presentation of the Racine stage 5 seizure. The following day, the fully kindled VEH-, LTG-, or LCM-CKM were challenged with one of two doses of either (**A**) LTG or (**B**) LCM and mean seizure score assessed at the time of peak effect of each ASM. * indicates significantly different from baseline seizure score, *p* < 0.05; **** indicates significantly different from baseline seizure score, *p* < 0.0001.

**Figure 3 ijms-24-04848-f003:**
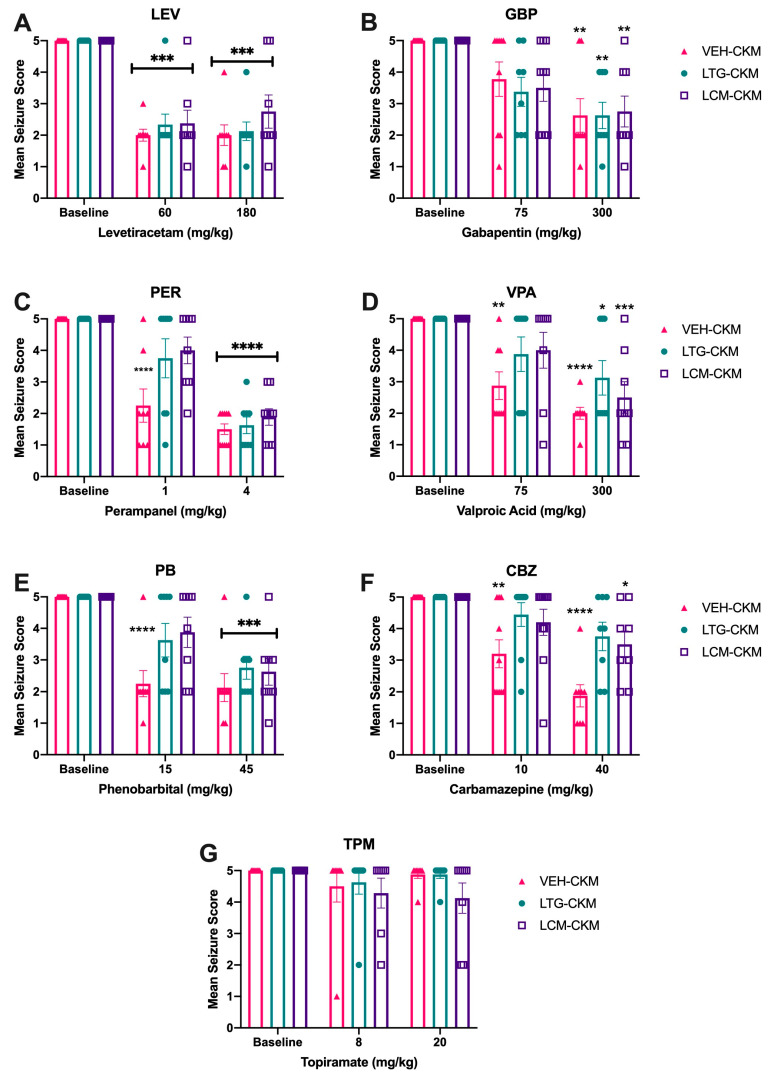
Repeated administration of either LTG or LCM to male CF-1 mice during 60 Hz corneal kindling leads to subsequent development of a drug-resistant acute secondarily generalized focal seizure model. Only escalating doses of the ASMs (**A**) levetiracetam (LEV) and (**B**) gabapentin (GBP) retained the same degree of anticonvulsant efficacy across VEH-, LTG- and LCM-kindled mice. Several ASMs lost potency in LTG- and LCM-kindled mice relative to that observed in VEH-kindled animals, including (**C**) perampanel (PER), (**D**) valproic acid (VPA), and (**E**) phenobarbital (PB). These agents were ineffective in LTG- and LCM-kindled mice at the low dose, whereas VEH-kindled mice demonstrated significant suppression of mean seizure score at this low dose. However, when the higher, nearly maximum-tolerated dose was acutely administered to LTG- and LCM-kindled mice, there was a significant reduction of mean seizure score, consistent with the effect of that same dose in VEH-kindled mice. The sodium channel-blocking ASM, (**F**) carbamazepine (CBZ), completely lost potency in LTG- and LCM-kindled mice at both low and high doses tested. Finally, (**G**) topiramate (TPM) was not effective at either dose tested in any kindling cohort. * indicates significantly different from baseline seizure score within kindling treatment group (* *p* < 0.05; ** *p* < 0.01; *** *p* < 0.001; **** *p* < 0.0001).

**Figure 4 ijms-24-04848-f004:**
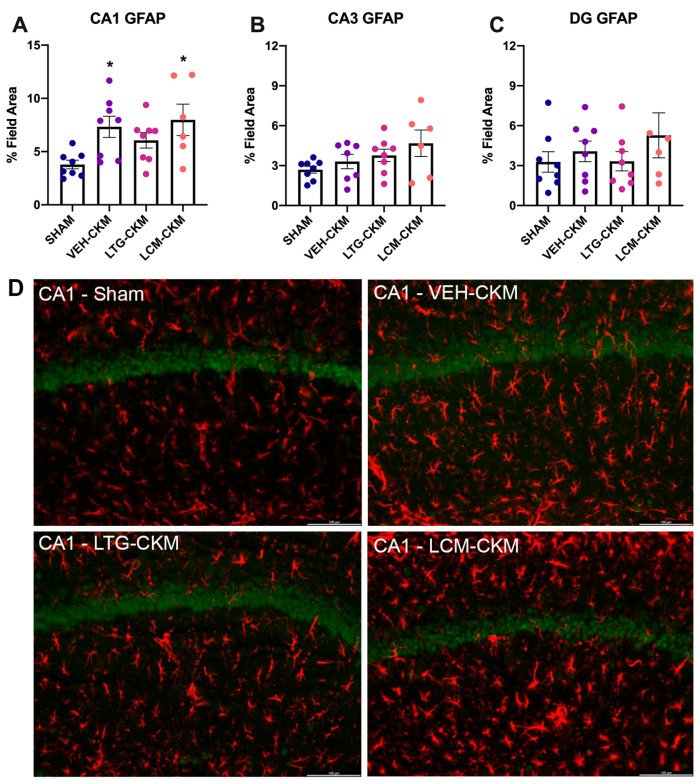
Administration of LTG (8.5 mg/kg) and LCM (4.5 mg/kg) during corneal kindling differentially impacts subsequent development of reactive astrogliosis in dorsal hippocampus. (**A**) There is increased GFAP immunoreactivity in area CA1 of dorsal hippocampus of VEH- and LCM-kindled mice versus SHAM-kindled mice, consistent with earlier findings in other CKM. There is no significant change in reactive gliosis in VEH- or LCM-kindled mice in either (**B**) area CA3 or (**C**) dentate gyrus (DG). (**A**–**C**) LTG-kindled mice do not demonstrate significant reactive gliosis in any region of dorsal hippocampus. (**D**) Representative photomicrographs of GFAP immunoreactivity in hippocampal CA1 demonstrate increased reactive gliosis in VEH- and LCM-kindled mice in this brain region. Red is GFAP; green is NeuN. Photomicrographs were captured with a fluorescent microscope (Leica DM-4) with a 20× objective (80× final magnification). * *p* < 0.05 relative to sham-kindled mice (*p* < 0.05).

**Figure 5 ijms-24-04848-f005:**
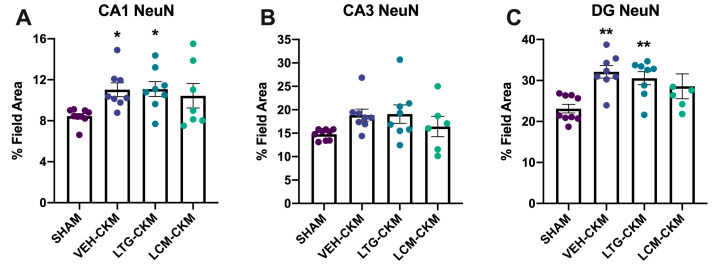
Neuronal density in VEH- and LTG- versus LCM- kindled mice was assessed in dorsal hippocampal structures one day after acquisition of the fully kindled state. Administration of LCM during the kindling process reduces chronic seizure-induced increases in NeuN immunoreactivity in hippocampal structures. (**A**) In area CA1, VEH-kindled mice and mice kindled in the presence of LTG exhibit significant increases in NeuN immunoreactivity (*p* < 0.05), an effect that is not observed in mice kindled in the presence of LCM. (**B**) There was no significant change in NeuN immunoreactivity in area CA3 in any group versus sham-kindled mice. (**C**) Chronic kindled seizures lead to increased NeuN immunoreactivity in dentate gyrus (DG) versus sham-kindled mice (*p* < 0.01). Mice kindled in the presence of LTG demonstrate similar chronic seizure-induced increases in NeuN immunoreactivity (*p* < 0.01), an effect that was not evident in LCM-kindled mice. * indicates significantly different from sham-kindled mice, *p* < 0.05. ** indicates significantly different from sham-kindled mice, *p* < 0.01.

**Figure 6 ijms-24-04848-f006:**
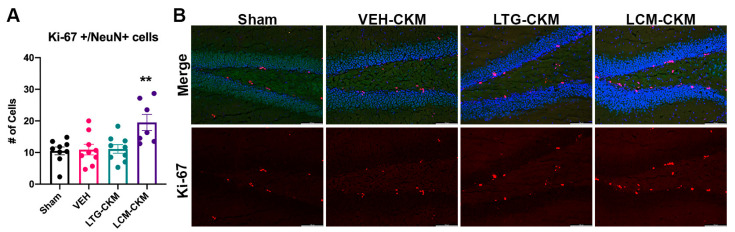
There is enhanced Ki-67 immunoreactivity in NeuN+ cells, presumed mature neurons, in the dentate gyrus of mice kindled in the presence of LCM relative to sham-kindled mice. (**A**) Neither VEH-kindled nor LTG-kindled mice demonstrate similar increases in Ki-67+/NeuN+ cell counts in this brain region. (**B**) Representative photomicrographs of Ki-67+ cells and merged Ki-67+NeuN+ immunoreactivity in dentate gyrus. Red indicates Ki-67-positive cells, green is NeuN, and blue is DAPI nuclear counterstain. Photomicrographs were captured with a fluorescent microscope (Leica DM-4) with a 20× objective (80× final magnification). ** indicates significantly different from sham-kindled mice, *p* < 0.01.

**Table 1 ijms-24-04848-t001:** LCM exerts dose-related anticonvulsant efficacy in seizure-naïve male CF-1 in the maximal electroshock (MES) test when administered 1 h prior to testing. Mice were challenged on the fixed speed rotarod immediately prior to MES stimulation to assess effects of LCM on minimal motor impairment. The degree of protection was assessed as the abolition of the hindlimb tonic-extension component of the MES-induced seizure. These data were used to quantify an i.p. median effective dose (ED50) of LCM (4.05 mg/kg [3.65–4.80]). Data are expressed as N = number protected in the MES test/F = number tested or T = number impaired on a rotarod/F = number tested.

Dose (mg/kg, i.p.)	Protected (N/F)	Motor Impairment (T/F)
2.5	1/8	0/8
3	1/16	0/16
4	5/15	0/15
5	8/8	0/8
10	8/8	0/8
15	8/8	0/8

**Table 2 ijms-24-04848-t002:** Repeated early exposure to anticonvulsant doses of sodium channel-blocking ASMs LTG (8.5 mg/kg) and LCM (4.5 mg/kg) during kindling leads to cross-resistance to both ASMs and dose escalation of the same ASM. Using a cross-over study design to assess ASM efficacy in male CF-1 mice previously kindled in the presence of LTG or LCM, a two-point dose-response evaluation was performed to determine the potential for resistance of fully kindled mice to various ASMs. Early LTG and LCM monotherapy promotes resistance to several other ASMs, except levetiracetam (LEV) and gabapentin (GBP). Gradient shading indicates degree of protection in greater than 50% of animals tested, with Racine stage 2 or lower seizure score considered “protected”. ^1^

ASM	Acute Low and High Doses Tested (mg/kg, i.p.)	VEH-Kindled Mice (N Protected/F Tested)		LTG-Kindled Mice (N Protected/F Tested)		LCM-Kindled Mice (N Protected/F Tested)	
		Low Dose	High Dose	Low Dose	High Dose	Low Dose	High Dose
LTG	17; 50	2/8	7/8	2/8	2/8	1/8	1/8
LCM	9; 26.5	3/8	4/8	0/8	3/8	2/8	3/8
LEV	60; 180	7/8	7/8	6/8	7/8	6/8	5/8
GBP	75; 300	2/8	6/8	3/8	5/8	3/8	5/8
PER	1; 4	6/8	8/8	3/8	7/8	1/8	6/8
VPA	75; 300	4/8	7/8	3/8	5/8	2/8	5/8
PB	15; 45	7/8	7/8	3/8	4/8	2/8	4/8
CBZ	10; 40	5/8	7/8	1/8	3/8	1/8	2/8
TPM	8; 20	1/8	0/8	1/8	0/8	1/7	2/8

^1^ Gradient shading indicates increasing proportion of mice protected from Racine stage 3 or greater behavioral seizure following administration of each ASM in the previously exposed kindling treatment groups. Red color indicates agents that were ineffective with protection ≤ 14.3%; orange color indicates agents that were ineffective with protection ≤ 25%; light green indicates moderately ineffective compounds with protection ≤ 38%. Effective agents are colored in white to increasing intensity of green (i.e., protection in ≥ 50% of animals tested). ASMs were delivered by the i.p. route and tested at the previously defined time of effect for each agent.

## Data Availability

The data presented in this study are available on request from the corresponding author and presented as individual datapoints in all accompanying tables and figures in this manuscript.

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
