# Peer review of "Frontline Sodium Channel-Blocking Antiseizure Medicine Use Promotes Future Onset of Drug-Resistant Chronic Seizures"

_ijms, 2023, doi:10.3390/ijms24054848_

Round 1

Reviewer 1 Report

Excellent manuscript. The study of the causes of drug resistance in patients with chronic seizures is absolutely necessary. The results of this study in a mouse model highlight the possibility that early and repeated administration of sodium channel-blocking ASMs promotes chronic drug-resistant seizures.

I would only ask for clarification on the criteria used to determine the low and high doses for each of the ASMs used (ie, recommended status epilepticus or maintenance dose).

Author Response

We are grateful to Reviewer 1 for this helpful feedback and positive review of our originally submitted manuscript. We have revised the manuscript to clearly explain the rationale for the ASM doses tested. The doses used were based on prior published median effective doses with the various ASMs in use in a variety of mouse seizure and epilepsy models and were meant to span a similar range of anticonvulsant efficacy male CF-1 mice. We have added this clarifying language in the manuscript body on page 4, lines 137-140.

“The doses selected for the various ASMs matched those that have been previously evaluated in the LTG-resistant corneal kindled mouse and previously published median effective doses for these agents in a diversity of mouse focal seizure models [4,8,20].”

Reviewer 2 Report

After reading this paper several times I still had difficulty in understanding details of the experimental procedures, and consequently of the results.

Probably the most important uncertainty related to the seizures that had occurred after two weeks of exposure to antiseizure medication, or vehicle, in the kindled mice. Were the seizures that occurred spontaneous ones, or were they provoked? If provoked, was the provocation the electrical stimulation method employed in kindling? If the latter, was there much chance that the various antiseizure medications investigated would have worked, particularly in the absence of evidence (as you noted) that potentially effective drug concentrations were attained, when two antiseizure medications had already been incapable of preventing kindling in the animals?

There seemed to be inconsistencies in the number of animals studied in different parts of the paper. The Materials and Methods section seemed to indicate that there were 40 kindled and 10 sham-kindled mice, yet in figure 1 there seemed to be a total of 63 mice, and in table 2 a total of 72 mice. I think making the logistics of the seizure production part of the methodology clearer would help the reader.

The title of your paper refers to drug-resistant chronic seizures, but did your studies continue long enough in individual animals after kindling was induced for you to be entitled to claim that the findings would apply in the long-term situation?

You seem to include phenobarbital among the sodium channel active drugs, and I think it is usually regarded as acting mainly on chloride channels in its human therapeutic use though sodium channels are also involved at higher dosages.

Your findings and the suggestions you make arising from them tend to be rather contrary to clinical experience in managing human focal epilepsy. You may well be correct, but I wonder if it might be wise to discuss how adequately your electrical kindling method reflects what probably happens when an abnormal focal discharge occurs in cerebral grey matter.

Author Response

Regarding Table 1, we apologize for this oversight! We have added the necessary methodological details to define the MES testing in seizure naïve mice, which was conducted prior to any LCM administration during the corneal kindling process. These MES test data were necessary to confirm that the dose of LCM to be used in the kindling procedure would match our prior study design with the MES ED50 dose of LTG (Koneval et al, 2018). We have added further details in the methods section beginning on page 14:

Maximal Electroshock Test (MES): A cohort of seizure-naïve mice (n=63; Table 1) were used to define the median effective dose (ED50) of LCM in male CF-1 mice from Envigo to confirm alignment with previous reports in CF-1 mice from Charles River [19] prior to embarking on any drug administration studies during the corneal kindling process. LCM was administered by the i.p. route 1 hour prior to MES testing. For the MES test, 60 Hz of alternating current was delivered for 2 seconds by corneal electrodes. An electrolyte solution containing an anesthetic agent was applied to the eyes before stim-ulation (0.5% tetracaine HCl). The current intensity was 50 mA for mice. A mouse was considered “protected” from MES-induced seizures in the absence of the hindlimb tonic extension component of the seizure [4,47-49].

We apologize for the unclear testing procedure related to the data represented in Table 2. These animals in Table 2 are the same 40 corneal kindled mice that began the study and animals were serially reused for dose-escalation studies with each of the candidate ASMs after exposure to LCM or LTG. We have further clarified this information throughout the revised text on page 15:

“The same mice that were initially exposed to VEH, LTG, or LCM during the corneal kindling process and determined to be fully kindled were then used in a cross-over ASM screening design for up to 2 months after attaining the kindling criterion, matching our previously reported study design and ASM testing procedure [8,20].”

This addition also addresses the Reviewer’s query related to the long-term study design of the monitoring. For clarity, animals were serially reused for up to 2 months after attaining the fully kindled state because the kindled seizure is stably evoked each day prior to acute ASM testing. These details have been fully and extensively previously detailed by our group and others (Koneval et al, 2018; Rowley and White 2010), thus we did not define these parameters in detail but instead included relevant references.

We apologize if there was previous misrepresentation of phenobarbital’s mechanism of action, which is most certainly not sodium channel modulation. We have confirmed that the revised text is clear on the MOA of phenobarbital.

Finally, we appreciate the Reviewer’s input on the clinical relevance of these findings and have revised the text body throughout to address this comment.

Reviewer 3 Report

This is an interesting article although the contents is complicated and not easily comprehensible especially for non-professionals. However, the paper should be accepted after a minor revision. Two points should be addressed:

1) The Authors should provide information about the specificity of the drugs for different types of voltage-gated sodium channels. Would it be possible that the development of drug-resistant chronic seizures is a consequence of a long-term up-regulation of drug-resistant Nav channels ?

2) Is there any information about side-effects of the drugs on voltage-gated Kv channels or inward rectifier Kir channels ? Would it be possible that the development of drug-resistant chronic seizures is a consequence of a long-term down-regulation of drug-snsaitive Kv or Kir channels ?

Author Response

  1. We appreciate this helpful feedback and comment from the Reviewer. We have further clarified the specificity of the various ASMs for voltage-gated sodium channels in the discussion. We do believe that there is some potential up-regulation of sodium channels and ongoing studies are addressing this more thoroughly. We have added further discussion of this potential phenomenon but agree that additional studies are clearly needed. New language has now been added on page 12:

“Instead, the present study demonstrates that ASM monotherapy alone can strikingly influence the resulting neuropathology, which warrants further detailed study. Volt-age-gated sodium channels may promote T-cell development [46], activation [47], and invasiveness [48], further suggesting that neuroinflammation may have been chronically differentially modulated by the LTG versus LCM administration that requires additional mechanistic studies.”

  1. As in response to Reviewer 3’s query #1 above, we also greatly appreciate this helpful comment. We have also added further detail to address this question but do not have any data at this point to addresses any changes in potassium channels. Future studies will most certainly address this point. New language has now also been added on page 12:

“It is also possible that chronic blockade of sodium channels can evoke functional changes in potassium channel function, as a similar phenomenon occurs following chronic ex-posure to the potassium channel blocker, dofetilide, in cardiomyocytes and leads to in-creased late sodium currents [42]. Thus, additional investigations into the bidirectional functional changes in sodium and potassium channel expression and function after chronic exposure to LCM and LTG during epileptogenesis is clearly necessary.”

Round 2

Reviewer 2 Report

It seems that the seizures assessed after 2 weeks of ASM exposure are electrically-induced and not spontaneous. I can appreciate the difficulties related to somehow recording spontaneous seizures in the experimental situation dscribed, but this does raise the question of the adequacy of the experimental preparation as a model of human chronic seizure disorders.